



# Determination of Heavy Rain Damage-Triggering Rainfall Criteria Based on Data Mining

Jongsung Kim[1], Donghyun Kim[1], Changhyun Choi[2], Myungjin Lee[1], Yonsoo Kim[3], Hung Soo Kim[1]

[1]Department of Civil Engineering, Inha University, Incheon, 22212, South Korea
[2] Risk Management Office, KB Claims Survey and Adjusting, 06212, Seoul, Korea
[3] RM Research institute, LIG System, 03130, Seoul, Korea

*Correspondence to*: Hung Soo Kim (sookim@inha.ac.kr)

**Abstract.** Heavy rainfall occurs over the Korean peninsula mainly because of typhoons and a localized heavy rainfall, leading to severe flooding and landslide risk. KMA (Korean Meteorological Administration) has the criteria for issuing a Heavy Rain

Advisory (HRA) over the peninsula even though each region or local government has different conditions in capability of disaster prevention (CDP) and different characteristics in rainfall and heavy rain damage. Therefore, the aim of this study is to suggest the methodology for the determination of Heavy rain Damage-Triggering Rainfall Criteria (HD-TRC) that HRA can be issued in each region. The study regions are local governments in Gyeonggi-province, Seoul-city, and Incheon-city in Korea. HD-TRC can be determined based on rainfall and heavy rain damage data. The data from 2005 to 2018 are collected and then

the data for flood or rainy season from June to September are extracted. The rainfall data is provided in KMA and heavy rain damage data during disaster periods (DPs) can be obtained from the statistical yearbook of natural disaster (SYND) published by MOIS (Minstry of Interior and Safety) every year. Training set of 2005 to 2014 is used for obtaining HD-TRC and test set of 2015 to 2018 is used for evaluating three criteria of HD-TRC, Advanced HD-TRC, and HRA. Analysis for determining the best criteria is performed through data mining processes as follows: (1) Maximum rainfalls in durations of 1 to 24-hr ($X_1$) and

antecedent rainfalls of 1 to 7-day ($X_2$) are obtained and used as independent variables. Heavy rain damage data are divided into damage day ("1") and no damage day ("0") used as dependent variables ($Y$). Principal component analysis (PCA) is performed and PCs (principal components) are obtained as $PC.X_1$ and $PC.X_2$ for independent variables. Then Risk Index (RI) is defined as $PC.X_1 + PC.X_2$ and RIs become the candidates for HD-TRC. The predicted damage($\hat{Y}$) is obtained based on RIs and confusion matrix is constructed then the best HD-TRC is determined through the evaluation of classification performance.

(2) However, 'abnormal days' (ADs) in a DP that the damage is occurred exists. The ADs mean the days which we do not have rainfall or have small rainfall amount during DP. Say, ADs have too small rainfall to damage even during DP. The ADs are defined as days below rainfall of 20mm and 5 cases of ADs are also defined as 0, 0~5, 0~10, 0~15, and 0~20mm in this study. We count total days in all the DPs and in ADs for a case. The ratio of ADs to total days during DPs could be the occurrence probability or prior probability (PP) of ADs for a case and 5 PPs are obtained. Also, the average AD for each case

can be obtained and defined as risk range (RR). Then we define Advanced HD-TRC using MCS (Monte Carlo Simulation) linked with PP, RR, and  from HD-TRC for each case. Therefore, HD-TRC is determined based on RI and Advanced HD-



TRC for each case based on PP and RR. Finally, three criteria of HD-TRC, Advanced HD-TRC, and HRA are compared based on performance evaluation by test set. As a result, Advanced HD-TRC shows the best performance and so the suggested methodology can be used for regional heavy rain damage warning information.

## 1 Introduction

The frequency of natural disaster and damage scale are increasing trend over the world due to climate change and rapid urbanization. In the last decade, total damage by natural disaster was 3.4 trillion KRW (Korean Won) (USD 2.83 billion) in Korea. Especially, the damage by heavy rain was 3.4 trillion KRW (USD 1.25 billion) (Ministry of the Interior and Safety, MOIS, 2018). Therefore, the structural measures such as dam, levee or dike, and flood control channel have been constructed while the nonstructural measures of natural or heavy rain disaster forecasting and warning system have been also used for the damage reduction. However, in recent, people are aware of the importance of the conservation of nature and the value of ecosystem and so these days, our preference is for the nonstructural measures rather than the structural measures. Forecasting and early warning system for flood or heavy rain damage is one of examples of the nonstructural measures which can give early warning to the affected people. This can provide information for preparing to match urgent needs with available resources and for emergency response according to an action plan then it can considerably reduce heavy rain loss and damage, and the loss of human lives.

However, the accuracy of forecasting and warning system is required in two aspects. First is about the accuracy of meteorological information which is forecasted by various models and the other is the accuracy of warning criteria. Weather forecasting has been studied for a long time and various models have been used for the future prospects of meteorological information considering climate change (Kannan et al., 2010; Abbot et al, 2012; Mekanik et al, 2013; Abbot et al., 2014; Kim et al., 2014). Many researches have been also performed for the evaluation of flood and drought risks by using the future meteorological information (Kay et al., 2006; Dankers et al., 2008; Robert et al., 2012; Zbigniew et al., 2013). But the accuracy of long-term forecasting for the future can not be guaranteed because many uncertainties are involved in observations and information and so the study has concentrated on short-term forecasting of meteorological information using various statistical methods such as regression model, neural network model, machine learning, and so on (Chau et al., 2010; El-Shafie et al., 2011; Abhishek et al., 2012; Kim et al., 2013;Cramer et al., 2017; Mishra et al., 2018; Chatterjee et al., 2018; Luitel et al., 2018). While the researchers have interested in the studies for meteorological forecasting, there have not been enough studies to develop the warning criteria.

The early warning system for landslide has warning criteria according to earthquake intensity but there is no warning criteria for heavy rainfall in Korea. Actually, the landslide and failure of slope are related to the localized heavy rainfall and typhoons mostly occurred in June to September. Therefore, the previous studies said that the warning criteria of landslide should be determined according to heavy rainfall and so the statistical, empirical, and satellite methods have been applied to determine the critical or threshold value of landslide-triggering heavy rainfall (Glade et al., 2000; Piciullo et al., 2016; Rossi et al., 2017;





Lee et al., 2017). Also, the early warning for flood is determined by threshold value of flood discharge obtained from flood
forecasting based on hydrologic rainfall-runoff model (Beguería et al., 2006; Montesarchio et al., 2009; Alfieri et al., 2015;
Forestieri et al., 2016; Miao et al., 2016; Sieq et al, 2017; Zhai et al., 2018; Sairam, 2019). However, the rainfall-runoff model
can be used just for a specific region around the river or a river basin. There are statistical methods of rainfall ensemble and
the relationship of IDF (Intensity-Duration-Frequency) curve and the return period for determination of critical rainfall (Kim
et al., 2011; Bezak et al., 2015; Yang et al., 2016). The critical rainfall for improvement of Heavy Rain Advisory (HRA) or
Heavy Rain Warning (HRW) is determined by the occurrence probability of damage in each rainfall duration obtained by
Bayesian theory (Park et al., 2014; Montesarchio et al., 2015; Song et al., 2016; Cho et al., 2018; Jeong et al., 2017; Lopez et
al, 2017; Choi et al., 2018; Lee et al., 2018; Kim et al., 2018). The previous studies have mostly suggested the criteria based
on the specific rainfall intensity and this can explain for the localized heavy rainall but it is difficult to explain for relatively
small amount of rainfall having long duration. In addition, the damage day was not considered in most of previous studies, and
among them, there was no study considering the uncertainty that damage could occur even in small rainfall.

Therefore, this study considers rainfall intensities of 24 durations and antecedent rainfalls of 1 to 7 days for explanation of
various rainfall events. Also this study consider the uncertainty of ADs additionally. The purposes of the study are to
understand the regional rainfall and heavy rain damage characteristics based on data mining and to suggest Heavy rain
Damage-Triggering Rainfall Criteria (HD-TRC) for each region or local government. Then we are going to propose Advanced
HD-TRC through considering uncertainty of ADs stochastically.

## 2. Heavy Rain Warning Criteria and Statistical Yearbook of Natural Disaster

### 2.1 Warning Criteria for Heavy Rain

Early warning system has been studied and utilized in most of the countries to mitigate natural disaster risk. Especially, heavy
rain warnings are issued with the criteria when heavy rainfall is expected to cause serious damage, such as severe flooding
and/or landslides. As can be seen in Table 1, heavy rain advisory and warning are announced by soil water index (SWI) and
heavy rainfall intensity of the best segmentalized administrative district by JAM (Japan Meteorological Agency) in Japan.
SWI is for the stored water quantity in soil and higher SWI describes higher disaster risk. Hong Kong Observatory (HKO) is
operating early warning system of Amber, Red, and Black based on heavy rainfall intensity. Canada (http://www.weather.gc.ca)
divided the region into three groups and is utilizing the early warning system for heavy rainfall. HRA and HRW are issued by
KMA when the forecasted heavy rainfall intensity is above the criteria.

**Table 1.**

Japan, Hong Kong, and Canada are considering the regional characteristics for HRA or HRW while the same criteria over the
Korean peninsular without consideration of local properties has been used for issuing HRA and HRW in Korea. In reality,
each local government or region has different Capability of Disaster Prevention (CDP). When the same criteria for HRA is





applied to A region having lower CDP and B region higher CDP, the damage may be occurred in A region but not in B region. That is to say, A region can have damage even though HRA is not issued but B region does not have damage even though HRA is issued. If the regional characteristics are not properly reflected for HRA, this causes people to lose the trust for the criteria of HRA. Therefore, we may need the regional criteria of HRA which can consider heavy rainfall, heavy rain damage,

and CDP characteristics of each local government. To do this, the methodology for derivation of Heavy rain Damage-Triggering Rainfall Criteria (HD-TRC) that means the critical or threshold rainfall criteria in each region or local government which the damage can be occurred are suggested in this study.

## 2.2 Statistical Yearbook of Natural Disaster

The SYND (statistical yearbook of natural disaster) which records the data related to the damage occurred due to natural disaster is published by MOIS (Korean Ministry of the Interior and Safety) every year from 1985 to present. It is categorizing the damage data according to meteorological hazards such as typhoon, heavy rainfall, heavy snowfall, strong wind, ocean wave and so on. Damage data is also organized in SYND for each region such as si (city), gun (county), and gu (district). MOIS defines Disaster Period(DP) in SYND as HRA period issued by KMA and the damage is recorded for casualty, victims,

public facility, and private facility during DP (see Table 2). When a rainfall event is moving over the regions, the event affects from a region that the first damage is occurred to the final damaged region and this moving period of an event is DP. However, there exists a certain region of no rainfall even during DP and so the damage could not be occurred in the region.

Table 2 is describing an example of heavy rain damage occurred from August 26, 2018 to September 1, 2018 for Nowon-district in Seoul city and this is recorded in SYND published in 2019. There are the damage items, DP in the top of right side,

and total damage in the bottom of right side in SYND. Total damage was 317.94 KRW in millions and KRW is KoRean Won (1USD=KRW/1,200). Therefore, all the damage and DP over the country are recorded in SYND and this study collected the heavy rain damage data and DP from 2005 to 2018 for local governments in Gyeonggi-province, Seoul-city, and Incheon-city (see a section 5.1).

**Table 2**.

The damage data and the damage days during DPs are collected for the study regions and Fig. 1 is showing the damage and damage days in all of the study regions (66 districts). The red line represents the no. of damage day and green bar is for the damage. Annual mean damage was about USD 38 million and the number of annual mean damage was 348. The biggest damage and damage days or damage frequency were occurred in 2011 and the damage frequency in 2006 is similar with 2011 but the damage was relatively smaller than that of 2011. The damages and damage frequencies in 2014 and 2015 were very

small and so we can know that it is describing drought years.

**Figure 1.**



## 2.3 Rainfall Data Collection

Rainfall data is observed and managed by KMA, MOLIT (Ministry of Land, Infrastructure and Transport), and K-Water in Korea. Especially, the relatively long and reliable rainfall data record can be obtained from KMA. ASOS (Automated Surface
Observing System) and AWS (Automatic Weather System) are used for measuring meteorological data including rainfall. The rainfall data has been observed from 1970s by ASOS which has 101 stations in South Korea while the data observed from 2000s by AWS which has 501 stations. Therefore, this study uses the record of AWS which has more densely distributed stations and Thiessen polygon method is used for spatial distribution of the rainfall. The hourly rainfall data by AWS is collected from 2005 to 2018 and it is transformed to mean aerial precipitation of each region. Finally, this study collects
maximum rainfalls in durations of 1 to 24-hr and antecedent rainfalls of 1 to 7-day from hourly rainfalls for the administrative districts of each city, county, and district in Gyeonggi-province, Seoul-city, and Incheon-city, Korea.

## 3. Analysis Methods for Determination of HD-TRC

### 3.1. Data Mining

Data mining is the process to find the patterns and characteristics in large data sets involving methods at intersection of statistics, machine learning, and database system. It is not a simple technique and also called as KDD (knowledge-discovery in databases) (Hand, 2007; Olson, 2008). The overall processes for data mining are explained as follows (Hastie, 2009; Frank, 2011).

#### (1) Data Extraction

First, it is necessary to set the purpose of solving the problem. Based on the purpose of doing the analysis and deriving data characteristics, the selection and collection of raw data will follow.

#### (2) Data Preprocessing

Data preprocessing is the second step in data mining technique. This is used to transform the raw data into a useful and efficient
format. The phrase "garbage in, garbage out" is particularly applicable to data mining projects. If one use an incorrect data, the obtained result will definitely be incorrect. Thus data preprocessing refers to the process of supplementing raw data such as filling the missing data and removing the redundant data.

#### (3) Data Transformation

In order to utilize a meaningful data analysis, deriving a representative characteristic or sample data is deemed important. Representative characteristic or sample data are converted from a preprocessed data for analysis.

#### (4) Data Analysis





Data mining can be divided into three categories according to its purpose. The first category is association analysis, while the
second category is classification. The last one is the regression prediction, wherein various analysis techniques are being used
such as artificial neural networks (ANN), decision trees, and random forests. Algorithms uniquely developed by researchers
can also be used for analysis

**(5) Evaluation and Interpretation**

The final step is to evaluate and interpret the analyzed results. Whether patterns and characteristics are suitable are judged
through this process.

In this study, we tried to solve the classification problem related to prediction. The researchers performed data mining to derive
the rain threshold which can forecast heavy rain damage from a large rainfall data. In the case of Step 4 of the data mining
procedure, the most common classification techniques such as decision trees and random forest are widely used. However, this
study proposes a new algorithm for deriving rain threshold for intuitive understanding.

**3.2 Monte Carlo simulation**

In general, deterministic model can find an exact solution, while stochastic model maybe can not. In this case, a series of
random numbers can be repeatedly generated and simulated to find the answer to the approximation (Kroese, 2014). This
method is called Monte Carlo simulation (MCS). MCS performs risk analysis by building models of possible results by
substituting a range of a probability distribution for any factor with inherent uncertainty. Thus, MCS is useful for modeling
with uncertainty or for problems with mathematically complex conditions (Nabian, 2017).
As an example, MCS    can be used as a process for calculating pi(the ratio of the circumference of a circle to its diameter).
When calculating the area of a circle represented by $x^2 + y^2 = 1$ , the circle is completely contained within a square space
with four in area. When a random number that satisfies this conditions ($-1 \leq x \leq 1$ and $-1 \leq y \leq 1$) is generated in this
space, the area of the circle can be roughly calculated by multiplying the ratio of the number of random numbers in the circle
by the total area of 4. If the number of randomly sampled random numbers increases, more accurate calculation results can be
obtained. However, there may be some errors due to the stochastic method. In this study, the MCS was is used to express the
uncertainty of Ads (abnormal days) which the researchers have referred to as days without rainfall or have small rainfall
amount during DP (Disaster period). Then, we examine the improvement of classification performance when the uncertainty
is considered for HD-TRC.





### 3.3 Principal Component Analysis

Principal component analysis is an analysis that derive new variables of small dimensions using a covariance matrix or a correlation matrix. The new variables are then called Principal Components and are not correlated with each other (Hotelling, 1933). In this study, the average vector for the set of independent variables $X = (X_1, X_2, \cdots, X_m)$ is designated as $\mu$ and the covariance matrix is $\Sigma$, with the diagonal element of $\Sigma$ is the variance $\sigma_1^2, \sigma_2^2, \cdots, \sigma_m^2$ of each independent variable. The number of eigenvalues of the covariance matrix $\Sigma$ is $m$, with descending order of eigenvalues called as $\lambda_1, \lambda_2, \cdots, \lambda_m$. Also eigenvectors

corresponding to each eigenvalue are called $e_1, e_2, \cdots, e_m$. The principal components are linear combinations with eigenvectors as coefficients (Wold et al., 1987). Covariance Matrix of principal component($Y = (Y_1, Y_2, \cdots, Y_m)$) are shown in Eq.(1) by characteristics of eigenvalue and eigenvector.

$$Cov(Y) = E^t \Sigma E = \Lambda, \Lambda = \begin{pmatrix} \lambda_1 & \cdots & 0 \\ \vdots & \ddots & \vdots \\ 0 & \cdots & \lambda_m \end{pmatrix} \qquad (1)$$

The eigenvalues are sorted by size, which means that some variances of the principal components may account for most of the

sum of the variances of all the variables. Thus the variance of the entire independent variable set $X = (X_1, X_2, \cdots, X_m)$ can be well explained by the variance of the principal components $Y_1, Y_2, \cdots, Y_k$ of the minority of $k$ (Jolliffe, 2011).

In this study, principal component analysis was considered for the following reasons. First is that the heavy rain damage may be caused by short-term heavy rains, while small amount of rainfall with long-term may continue to cause damage. This study considers all of rainfall variables including the maximum rainfalls in durations of 1 to 24hr ($X_1$ ) and the antecedent rainfalls

of 1 to 7days ($X_2$) in order to explain the damage caused by various types of rainfall events. However, because this study considers many rainfall variables, it is difficult to determine the criteria as a representative rainfall threshold. Therefore, we tries to reduce all the variables into one principal component ($PC$) which can represent both types of rainfall variables. Then, HD-TRC can be defined using $PC$s.

### 3.4 Evaluation of Classification Performance

This study uses "1" for heavy rain damage days, and "0" for no damage days. In order to evaluate the performance in classification problem, the confusion matrix should be calculated (Fawcett, 2006). The confusion matrix is explained in Table 3 which has Positive(P) for "1" and Negative(N) for "0". If the predicted value($\hat{Y}$ ) is 1 and the observed value($Y$ ) is 1, it is called as True Positive(TP), and if $\hat{Y}$ is 0 and $Y$ is 0, it is True Negative(TN). In addition, if $\hat{Y}$ is 1 and $Y$ is 0, it is False Positive(FP), and if $\hat{Y}$ is 0 and $Y$ is 1, it is False Negative(FN).

**Table 3**.

The performance evaluation indicators through confusion matrix can be defined and calculated (Table 4). These indicators are Accuracy, Error Rate, Sensitivity, Precision, and Specificity then, these can be used according to the purpose of the analysis (Powers, 2011).

**Table 4**.





There are 10,000 data. Say, P ("1") is 1,000 and N ("0") is 9,000. This has imbalanced data structure. If we predict 10,000 data as 0, accuracy is 90%. because N is 9,000. However, sensitivity is 0%. Therefore, it is difficult to evaluate classification performance when one indicator is used and so combinations of indicators such as AUC, ROC curve, and F1-score should be used.

The ROC (Receiver operating characteristics) curve is created by plotting the Sensitivity against the 1-Specificity at various

threshold settings. The ROC curve is a graphical plot that illustrates the diagnostic ability of a binary classifier system as its discrimination threshold is varied. The area under the curve (AUC) of the ROC curve is the criterion for verification. Fig. 2 shows the concept of ROC curve and AUC. The AUC values are varied from 0.5 to 1 and closer to 1 means better classification performance. If we see Fig 2, the yellow polygon with AUC=0.95 has bigger area than the blue polygon with AUC=0.50. Due to its greater advantage, AUC is widely used as a representative performance evaluation indicator for comparing the absolute

predictive performance (Choi, 2018). In general, it is appropriate to use AUC for a performance indicator of a balanced data. However, when the data is imbalanced, it is difficult to evaluate the performance correctly using AUC.

F1-score is a performance evaluation indicator that can be more accurately evaluated when the data is unbalanced in binary classification. As indicated by Sasaki, 2007, F1 score shown in Eq. 2, can be calculated through the harmonic mean of the precision and recall.

**Figure 2**.

$$\text{F} - \text{score} = \frac{(1+\beta^2)(Precision \times Sensitivity)}{(\beta^2 Precision + Sensitivity)} \qquad (2)$$

$\beta$ is generally defined as 1 in F-Score, and thus, it is called F1-Score. In order to verify if F1-score is appropriate as a performance measure in the imbalanced data, Saito (2015) applied several evaluation indicators in both balanced data and

imbalanced data. As a result, most performance measure have shown equal performance in the balanced data (P : 1,000, N : 1,000) and imbalanced data (P : 1000, N : 10,000). However it was identified that the classification performance of Precision-Recall Curve has changed in imbalanced data. Therefore, this study uses F1-score which includes precision and recall as the performance evaluation indicator.

**4. Analysis Procedure for Determination of HD-TRC**

**4.1 Analysis Procedure for HD-TRC**

**4.1.1 Step 1 : Dependent and independent variables**

Heavy rain damage and rainfall data are collected from MOIS and KMA then dependent and independent variables are determined as shown in Fig. 3 and Table 5. The procedure in Fig. 3 is explained as follows:




① Daily damage data obtained during DP is divided into two groups of damage day and no damage day. Damage day is represented as 1 and no damage day as 0. These 1 and 0 become dependent variable, $Y$ (also, see Fig. 8).

② Hourly rainfall data collected from KMA is transformed to mean aerial rainfall by Thiessen polygon method. Then, maximum rainfalls in durations of 1 to 24-hr($X_1$) and antecedent rainfalls of 1 to 7-day($X_2$) are obtained and these become independent variables of $X_1$ and $X_2$.

③ Thus, we establish dependent variable $Y$ and independent variables $X = X_1 + X_2$. Also, the variables are divided into training set and test set.

<p style="text-align:center">Figure 3.</p>
<p style="text-align:center">Table 5.</p>

### 4.1.2 Step 2 : Rain index and performance test

The best HD-TRC is determined in this step by performance test for training set. Its procedure is explained in Fig. 4 according to following steps:

① The variables of $Y$ and $X$ for rainy season of June to September are extracted through preprocessing of the data

② Principal component analysis (PCA) is performed for $X_1$ and $X_2$ then one principal component (PC) for $X_1$ and one for $X_2$ are obtained as $PC.X_1$ and $PC.X_2$. The sum of two PCs are defined as Rain Index (RI), say, RI $= PC.X_1 + PC.X_2$.

③ All the RIs in training set become the candidates of HD-TRC. If RI at the point of time we want to predict is greater than RI which is a candidate of HD-TRC, we predict the damage will be occurred. Then the confusion matrix for Obs. ($Y$) and Pred. ($\hat{Y}$) is constructed and the classification performance test is conducted by F1-score for the evaluation of RI and the determination of HD-TRC. The RI having the best performance is determined as HD-TRC of the target region.

<p style="text-align:center">Figure 4.</p>


### 4.2 Analysis Procedure for Advanced HD-TRC

DP is defined as the period that heavy rain damage is occurred in SYND, that is to say, the period of HRA or HRW. Abnormal days(ADs) is also defined as the days which we do not have rainfall or have small rainfall amount or depth during DP(see ① and ④ of Fig. 5). If ADs are included during DP, the prediction error of damage occurrence can be leaded. Therefore, this

section defines Advanced HD-TRC considering uncertainty of ADs as explained in Fig. 5 as follows:

①, ②, ③ ADs of 5 cases are defined according to rainfall depth from 0 to 20mm during DPs. ADs of Case 1 is for no rainfall days during DPs, Case 2 is for rainfall days of 0~5mm, and Case 3 ~ 5 for rainfall days of each rainfall depth in ①. The





number of $N$ ADs during DPs is counted for each case and total number of days for all DPs ($M$) is also counted. Then the
ratio of $N/M$ is defined as Prior Probability (PP).

④, ⑤ ADs for the same DPs of each case are obtained and ADs in DPs are averaged for each case. The averaged AD for each
case can be rounded as an integer value. This averaged AD value is defined as Risk Range (RR). Here, PP is the occurrence
probability of AD and RR is the occurrence range of AD.

⑥ We can know DP if the heavy rain damage was already occurred but we can not know DP at the starting point of time for
the damage prediction. Therefore, the damage is predicted by assuming the damage will be occurred when the bigger
rainfall than HD-TRC is produced and we consider the probability and the range that AD is included. The random number
which is the probability is generated for the days corresponding to RR. The generated probability by MSC is compared
with PP then the predicted damage occurrence is determined (also, see a section 5.4 and Fig. 14). Therefore, Advanced
HD-TRC is defined according to PP and RR estimated in each Case.

290                                         **Figure 5**.

## 4.3 Comparative Analysis for Determination of the Best Criteria

This section evaluates three criteria of HD-TRC, Advanced HD-TRC, and HRA and the best one is selected. the evaluation
procedure is explained as follows (Fig. 6).

① HD-TRC, Advanced HD-TRC, and Heavy rain advisory(HRA) are applied to test set and we calculate performance
indicators such as F1-Score and AUROC by confusion matrix.

② Then the best criteria having the best performance is determined.

                                         **Figure 6**.

## 5. Determination of Heavy Rain Damage-Triggering Rainfall Criteria

**5.1 Study Region**

The methodology suggested in this study is applied to the local governments of the metropolitan area. Say, Gyeonggi-province
has 25 local governments or districts, Seoul-city 31, and Incheon-city 10 (Fig. 7). The study investigated the damage history
for each region and district then obtained the frequency and the average frequency of heavy rain damage for each region and
administrative district (Table 6). However, the damage history is not enough to suggest HD-TRCs for whole local governments
or districts of city, county, and district. Therefore, this study combine all the data of local governments for Gyeonggi-province,
Seoul-city, and Incheon-city and suggests HD-TRCs for three regions.

                                         **Table 6**.
                                         **Figure 7**.




## 5.2 Data Preparation

### 5.2.1 Dependent variable

As mentioned in a section 4.1.1, daily heavy rain damage data during DP is obtained by SYND from 2005 to 2018. Then the dependent variable, Y of "1" and "0" for damage day and no damage day is obtained and this is explained in Fig. 8. Blue color represents damage days of DP and other days has no damage.

**Figure 8**.

### 5.2.2 Independent variable

There are 66 administrative districts and 103 stations of AWS in three regions of Gyeonggi-province, Seoul-city, and Incheon-city. About 4.7 stations affect the each district and hourly rainfalls from 103 stations are collected from 2005 to 2018. Fig. 9 shows Thiessen polygon for the estimation of mean aerial rainfall and daily rainfall variable is obtained for each district. Then, the independent variables of maximum rainfalls in durations of 1 to 24-hr ($X_1$) and antecedent rainfalls of 1 to 7-day ($X_2$) are 320 obtained and the variables are grouped into two sets of training set (2005 to 2014) and test set (2015 to 2018).

**Figure 9**.

## 5.3 Determination of the best HD-TRC by performance test for training set

### 5.3.1 Data preprocessing


Annual rainfall and the number of damage day are compared for examination of similarity between them in Fig. 10 which is plotted by standardized values because of scale difference. There are drought periods in 2014 to 2015. Especially, there was an extreme flood period in 2011 and an extreme drought period in 2014 to 2015. However, in 2006, annual rainfall was generally in average value while the number of day showed relatively higher value. This means that the small rainfall duration 330 lasted long even annual rainfall was average value and the rain damage was frequently occurred. Therefore, mostly, the heavy rain damage is increased as the rainfall is increased and so understanding damage occurrence based on the heavy rainfall is needed for HD-TRC.

**Figure 10**.

Because the number of damage days are much smaller than that of no damage days, it is very difficult to understand the proper HD-TRC by using all the damage data. The study examines the monthly damage history to see the damage period during 2005 to 2018 and finds the damage has been occurred mainly during rainy days in rainy season of June to September (Fig. 11). Therefore, we collect the data for rainy season and determine HD-TRC by using collected data and suggested methodology.

**Figure 11**.





**Table 7.**

As we can see in Table 7, the number of damage and no damage days for rainy season are smaller than that for whole years (2005 to 2014). Especially, the number of no damage days "0" is greatly reduced but "1" is a little reduced. Say, the number of "1" is reduced by 15% but "0" which we do not need by 90% and so if we just consider rainy season the analysis becomes 345 more easier.

### 5.3.2 Rain Index

There are too many independent variables to suggest them as HD-TRC and so principal component analysis (PCA) is used for deriving the principal components of $PC.X_1$ from maximum rainfalls of 1 to 24-hr duration and of $PC.X_2$ from antecedent 350 rainfalls of 1 to 7-day. Then rain index (RI) is defined as following equation of (3).

$$PC.X_1 + PC.X_2 = RI \qquad (3)$$

In general, when the number of principal component is determined, the cumulative variance of each variable should be over 355 80% for minimizing information loss and reducing the variables. As a result, we identify the cumulative variances of $PC.X_1$ and $PC.X_2$ are over 80% (Table 8) and this means there is no problem to reduce the variables as two principal components. That is to say, $PC.X_1$ and $PC.X_2$ are estimated and RIs are also obtained for each region. In training set, all RIs can be used for examining daily damage occurrence and regarded as the candidates for the determination of HD-TRC.

**Table 8.**


### 5.3.3 Determination of HD-TRC by Rain Index

The daily damage data in training set is classified as "0" and "1" for the construction of confusion matrix and evaluation indices in Table 4 are calculated for the classification performance. For an example, RI 1 becomes a criteria candidate and other RIs are compared to RI 1 then the results of comparison will be the predicted damage occurrence ($\hat{Y}$). Next RI 2 will be another 365 criteria candidate then others are compared to RI 2 for obtaining the other predicted damage occurrence. In this way, the predicted damage occurrence is estimated and then the confusion matrix is obtained based on $Y$ and $\hat{Y}$.
The evaluation indices such as Sensitivity, Specificity, and Precision are calculated from the confusion matrix.

**Figure 12**






and then the performance measures of F1-score by Sensitivity and Precision, and AUC by Sensitivity and Specificity are estimated. There are the number of data for "1" and "0" in each region. Total epoch of Gyeonggi-province has 18,055, Seoul-city 13,295, and Incheon-city 5,069. Here, epoch means that all the criteria candidates are evaluated as much as the number of data. The criteria candidate which has the largest F1-score becomes HD-TRC. Fig. 12 is showing the F1-score in all of epoch

on each region. Also the red points are expressed criteria candidates which have the largest F1-score. Therefore, HD-TRCs are determined based on F1-score for the regions (Table 9).

**Table 9.**

Fig. 13 is showing RIs, the determined HD-TRC, and the predicted damage occurrence on training set for each region. Red is

for observed damage day, blue is for observed no damage day, triangle is for predicted damage day, and cross is for predicted no damage day. Therefore, the classification can be obtained. That is, red triangle is for observed and predicted damage days (TP), blue triangle is for observed no damage day and predicted damage day (FP), red cross is for observed damage day and predicted no damage day (FN), and blue cross is for observed and predicted no damage days (TN). The black line is for the determined HD-TRC for each region. The confusion matrix is constructed by the classification results (Table 10).

385                                                      **Figure 13**

RIs above the line of a determined HD-TRC in Figure 14 mean the predicted damage occurrence represented by triangle and RIs below the HD-TRC mean the predicted no damage occurrence by cross. The HD-TRC for each region is determined based on the best classification performance which has maximum F1-score. However, if we see Table 10, there are still the number

of FN and FP, and also it can be seen in red cross(FN) and blue triangle(FP). FN still includes ADs and FP includes no damage days even though heavy rainfall was occurred. Therefore, this study considers ADs in FN to improve the predictability or classification performance and analyzes ADs in a section 5.4. The improved HD-TRC is called Advanced HD-TRC. However, FP is not reconsidered here. The reasons why no damage, even during heavy rainfall, exists are different regional characteristics such as CDP, urbanization, resilience, and so on.

395                                                   **Table 10.**

## 5.4 Determination of Advanced HD-TRC by Risk Range

This study defines ADs as below daily rainfall depth of 20 mm during DP and 5 cases as 0, 0 to 5, 0 to 10, 0 to 15, and 0 to 20mm in Table 11. Total number of days for DPs (M) and Total number of Ads (N) are counted (see ② and ④ of Fig. 5).

Then prior probability (PP) and risk range (RR) are also defined in Table 11 and Figure 6. PP is the probability that AD is occurred and RR is the range that AD is occurred as explained in a section 4.2. We use all the raw data without preprocessing to identify more exact PP and RR in training set in this section while we used the data of June to September with preprocessing for HD-TRC in section 5.3.1.





**Table 11.**


Actually we do not know AD and DP for the future but we know PP and RR from observations. Therefore the study applies MCS for probabilistically obtaining the final predicted damage occurrence, $\hat{Y}$ by using PP and RR. An example is described for the case 2 in Table 11. Case 2 is for daily rainfall of 0 to 5mm, for PP = 0.3198(about 0.32), and RR = 2days obtained from training set. Fig. 14 is analysis procedure for an example. The first column is for row number, and second is for the damage

occurrence, $\hat{Y}$ which can be predicted by the comparison of RI from test set with the determined HD-TRC in training set. Third is for the generated probability(P) which AD can be occurred in test set and AD is corresponding to RR. The last column is for the final predicted damage occurrence, $\hat{Y'}$ obtained by the comparison of PP and P.

If we see Fig. 14, two predicted damage days, $\hat{Y}$ can be found in the row numbers of 4 (predicted damage day(1)) and 6 (predicted damage day(2)) as $\hat{Y}$ =1. RR is 2 days and this is regarded as $\pm 2$ days around $\hat{Y}$ =1. Therefore, RR(1) and RR(2)

can be considered as the risk range based on predicted damage day(1) of $\hat{Y}$=1 and also RR(2) and RR(3) based on predicted damage day(2). The random numbers are generated for the days corresponding to RR and this becomes the probability(P) that AD is occurred. However, RR(2) in predicted damage day(1) is overlapped with predicted damage day(2) and so the random number of 0.24 is produced just for one day. That is, P is generated and compared with PP for the determination of $\hat{Y'}$. If we see Figure 15, since the row numbers of 1 and 9 are not included in RR, $\hat{Y'}$ becomes 0. The generated random numbers for the

row numbers of 2, 5, 7, and 8 are less than PP=0.32 and so $\hat{Y}$=0 becomes $\hat{Y'}$=1. In the row 3, since P=0.84 is greater than PP=0.32, $\hat{Y'}$ =0. $\hat{Y}$ is 1 in the rows of 4 and 6 and so $\hat{Y'}$maintains 1. In this way, the analysis is performed for 5 cases and this procedure is defined as Advanced HD-TRC. That is to say, Advanced HD-TRC considers uncertainty of AD by applying MCS and obtains more improved $\hat{Y'}$.

**Figure 14**


## 5.5 Determination of the Best Criteria by the Performance Evaluation

The HD-TRC and Advanced HD-TRC for each Case were determined in training set (2005 ~ 2014). Also, these criteria were evaluated in test set (2015 ~ 2018) then the best criteria was obtained. All the results including HRA are tabulated in tables 12 to 14. As the results, HD-TRC shows much better than HRA of KMA in its performance. If we see F1-score to compare HD-

TRC with HRA, the performance of HD-TRC on Gyeonggi-province is improved 23.17%, Seoul-city 5.06%, and Incheon-city 19.37%. F1-scores of Advanced HD-TRCs for 5 Cases show better than HRA and HD-TRC. Advanced HD-TRC for Case 4 has the best performance in Gyeonggi-province and Incheon-city, and one for Case 5 has in Seoul-city. Therefore, the proper best criteria for warning and forecasting of heavy rain damage will be Advanced HD-TRC 4 in Gyeonggi-province and Incheon-city, and Advanced HD-TRC 5 in Seoul-city.

435                                                        **Table 12.**



**Table 13.**

**Table 14.**

## 6. Summary and conclusions

We have suggested a methodology for the derivation of the rainfall criteria which heavy rain damage can occur. To do this, data mining process was used and the performance of the derived rainfall criteria was evaluated systematically. Initial step includes defining Rain Index (RI) by applying principal component analysis for various rainfall variables and all RIs were considered as candidates of HD-TRC. The performance evaluation for RIs based on heavy rain damage data was carried out and the best HD-TRC was selected. However, even though we do not have rainfall or have small rainfall amount during DP,

'abnormal days'(ADs) that the damage occurs exist. Therefore this study considered HD-TRC and the occurrence probability of ADs using Monte Carlo simulation. Then we defined this process as Advanced HD-TRC. Then, three criteria of HRA of KMA, HD-TRC, and Advanced HD-TRC were compared through classification performance and the main results of this study are summarized as follows:

(i) The best HD-TRCs in regions showed a value of 5.030 in Gyeonggi-province, 10.881 in Seoul-city, and 5.109 in Incheon-

city.

(ii) The performance of HD-TRC by F1-score has improved compared to the HRA by 23.17% in Gyeonggi-province, 5.06% in Seoul-city, and 19.37% in Incheon-city.

(iii) The performance of the 5 Advanced HD-TRCs by F1-score has improved compared to the performance of HD-TRC for each region.

(iv) The best criteria for each region were selected as Advanced HD-TRC 4 in both Gyeonggi-province and Incheon-city, and Advanced HD-TRC 5 in Seoul-city.

(v) The performance of the best criteria for each region was evaluated by F1-score and AUC. The results are as follows:
    F1-score showed 37.94% in Gyeonggi-province, 30.04% in Seoul-city, 40% in Incheon-city.
    AUC showed 73.36% in Gyeonggi-province, 73.40% in Seoul-city, and 76.74% in Incheon-city.


Upon calculating the F1-scores, HD-TRC showed an improvement of 15% on average compared to HRA of KMA in all regions, while Advanced HD-TRC showed a 21% improvement. In addition, if we see evaluation results by AUC, HD-TRC was improved by 7.8% compared to HRA, while the Advanced HD-TRC improved by 20%. F1-score, which is known to be suitable for imbalanced data, was used as a performance measure. However, F1-score is a relative evaluation indicator, that is, it

depends on the degree of imbalance in the data. Therefore, AUC, which is an absolute evaluation indicator was also used in this study.

It was identified that the HD-TRC of Seoul City is the highest and the HD-TRC of Gyeonggi Province is the lowest. This signifies that the region with a high HD-TRC means that the damage can occur on relatively high rainfall intensity. It could be



judged that roughly, the CDP of Seoul City is higher than that of Incheon City and Gyeonggi Province. In this study, since

only the rainfall and damage data were used, the relative CDP could be roughly estimated. However if additional factors such as socio-economic factors or disaster prevention projects in future research are to be considered, it could be expected that the CDP in region can be explicitly identified. If the results of this study would be used for an early warning system, it is possible to support the disaster managers through predicting the occurrence of heavy rain damage. Thus, it is expected that the suggested methodology will contribute in establishing safe environment of our life from natural disaster such as heavy rainfall.


**Acknowledgement**

This research was supported by a grant(2018-MOIS31-009) from Fundamental Technology Development Program for Extreme Disaster Response funded by Korean Ministry of Interior and Safety(MOIS).

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

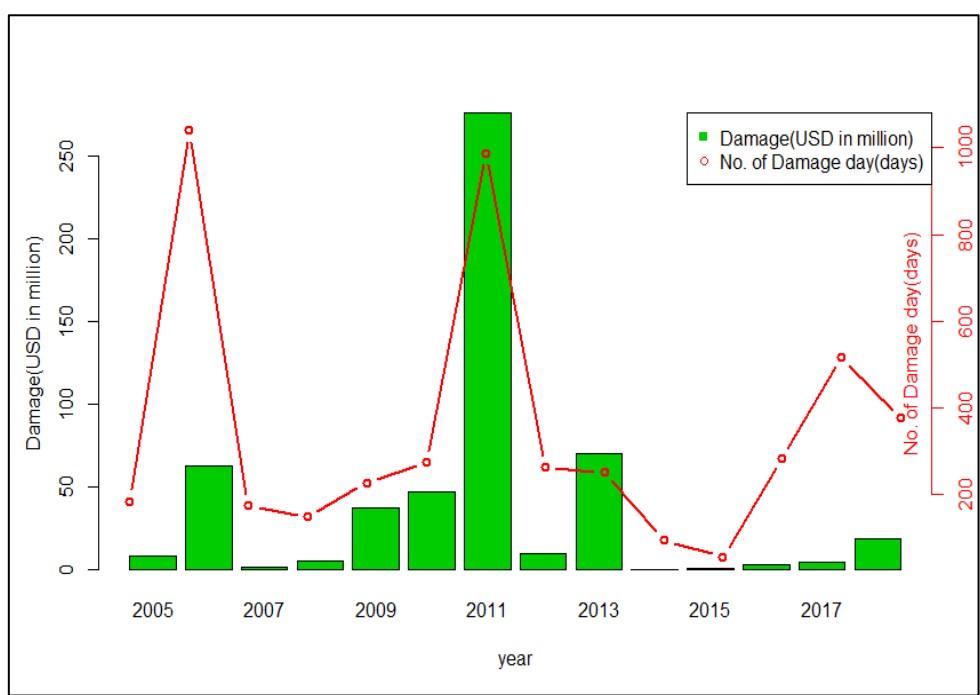


**Figure 1: Heavy rain damage and no. of damage days**

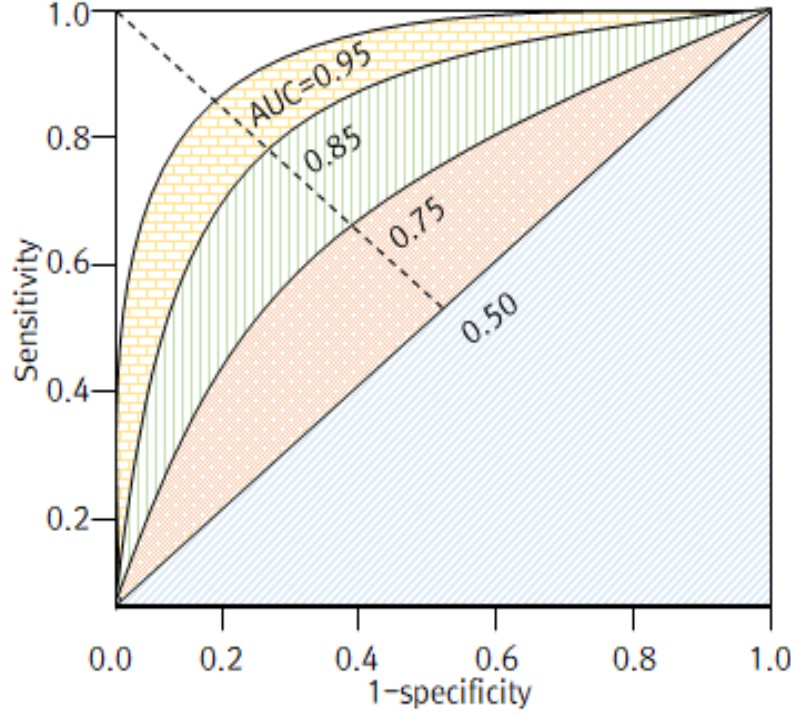

**Figure 2: Concept of ROC curve and AUC(Choi, 2018)**




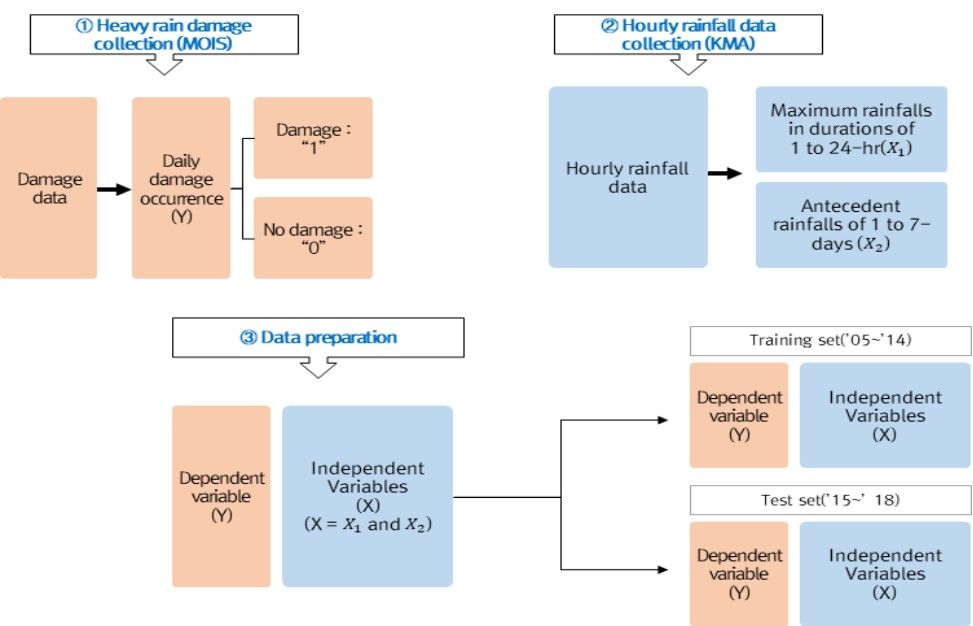

620           **Figure 3: Establishment of dependent and independent variables for HD-TRC**

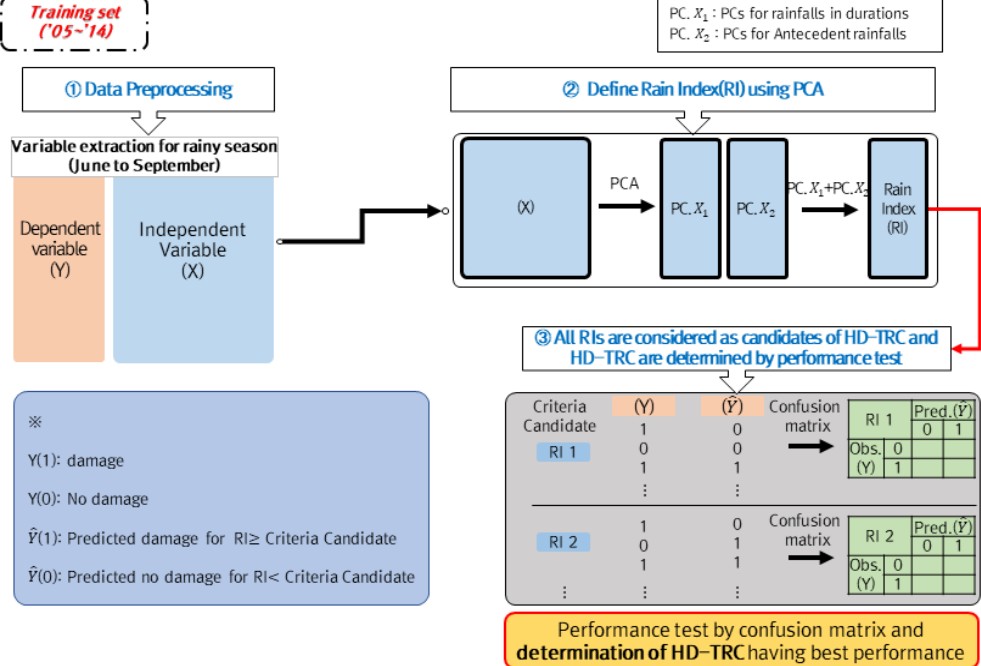

**Figure 4: Determination of the best HD-TRC by performance test for training set**




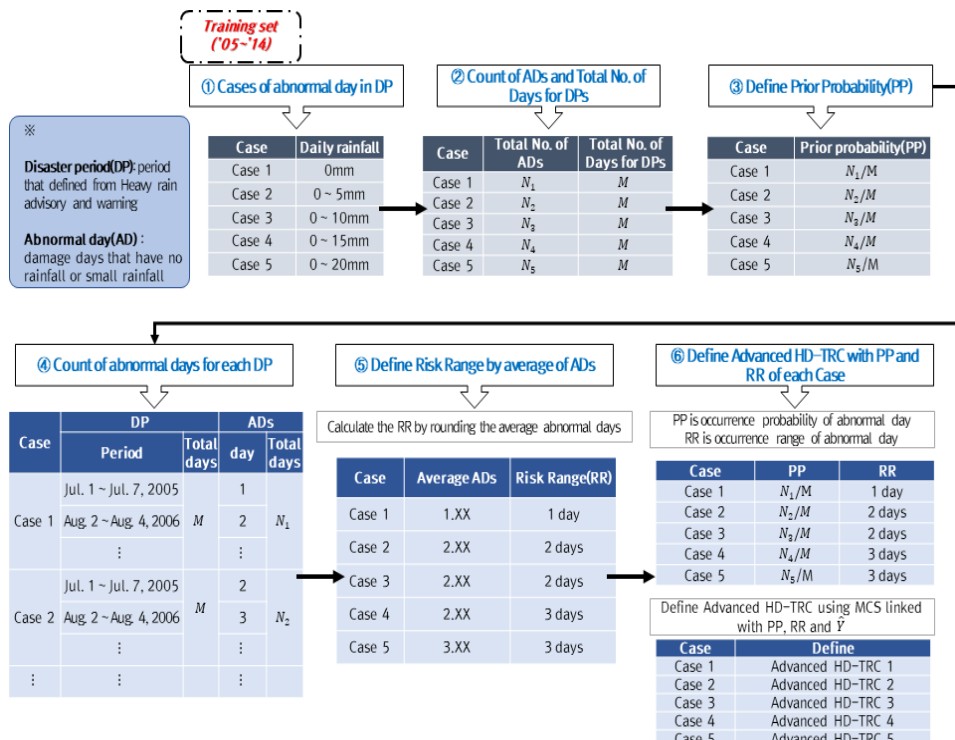

625            **Figure 5: Process of Advanced HD-TRC using occurrence probability and range of abnormal day**

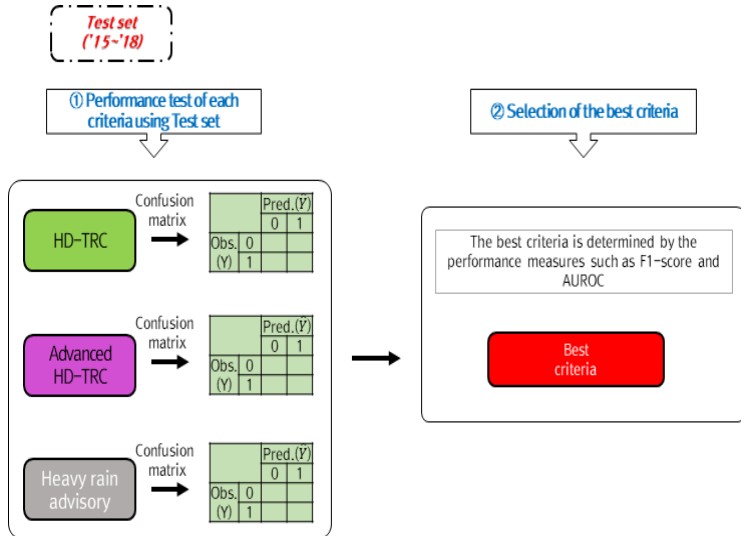

**Figure 6: Selection of the best criteria after comparison of performance**



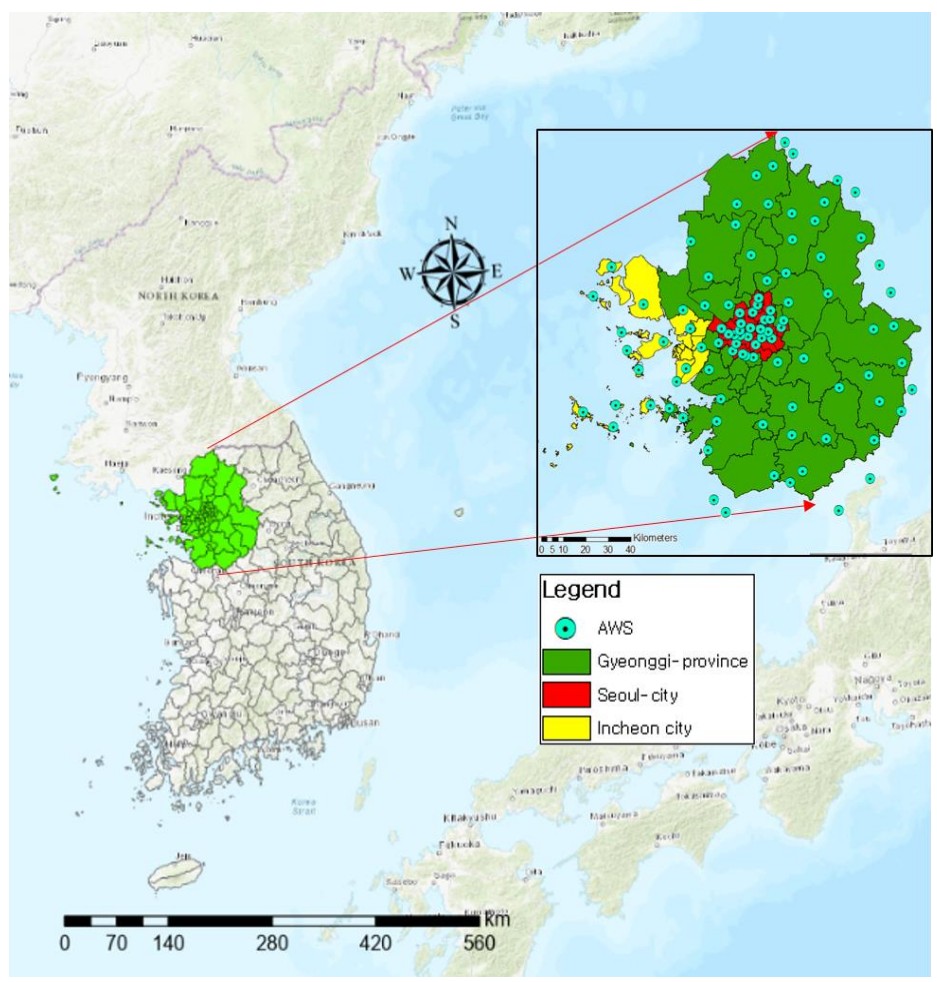

630                      **Figure 7: Study Region in the Korean peninsular**


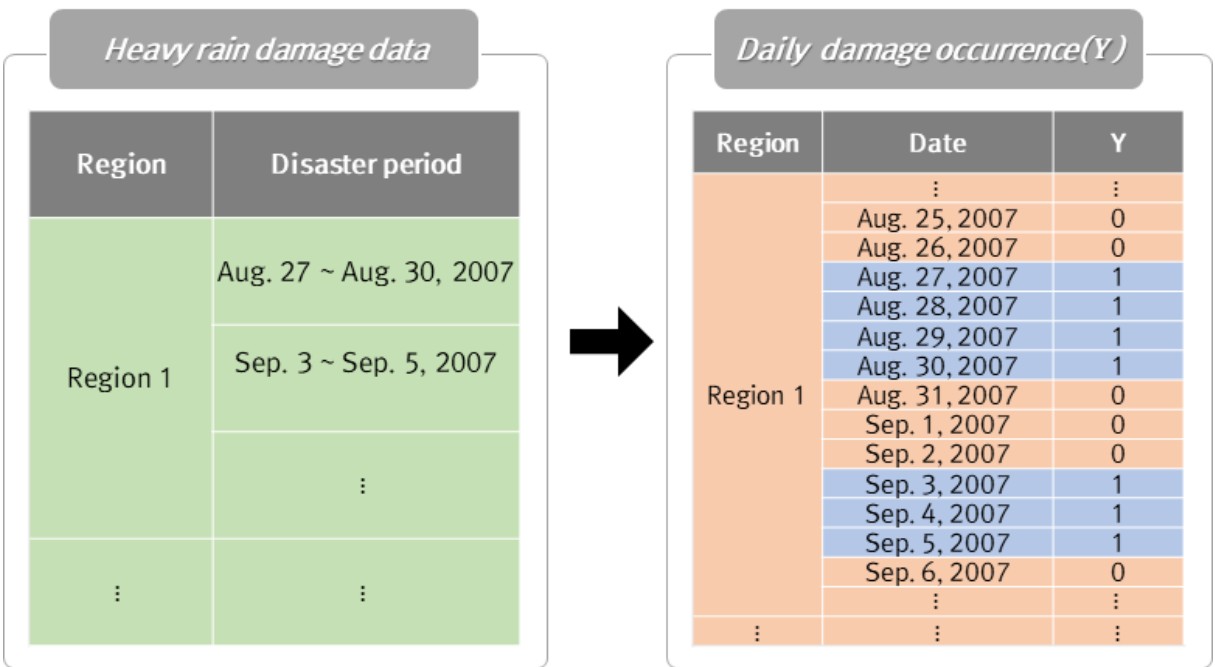

**Figure 8 Concept of daily damage occurrence to obtain dependent variable**

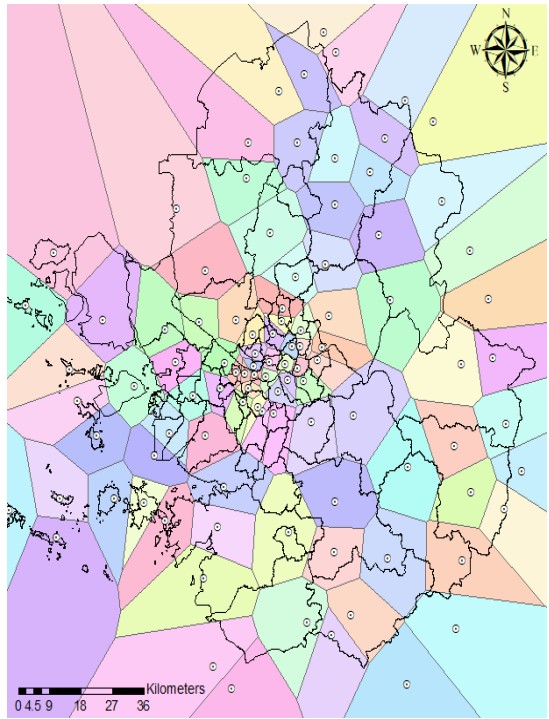

**Figure 9: Thiessen polygons for the study region**


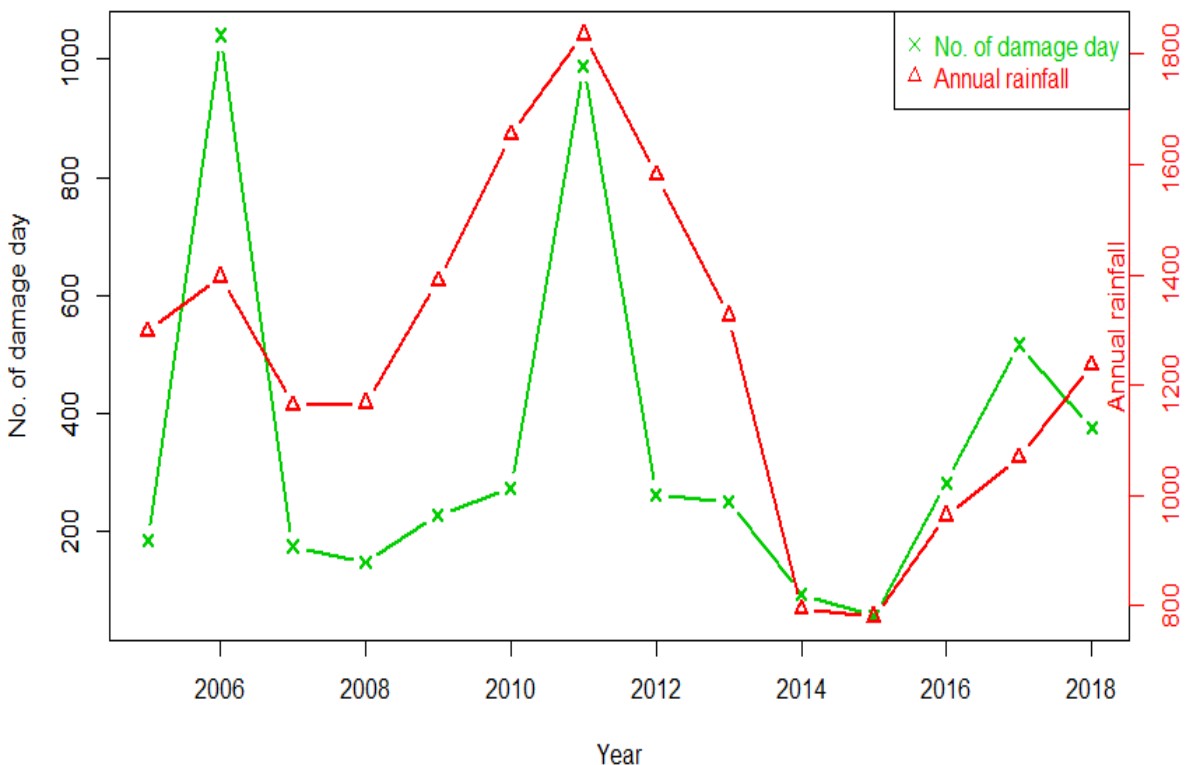

**Figure 10: Plots for number of damage day and annual rainfall**

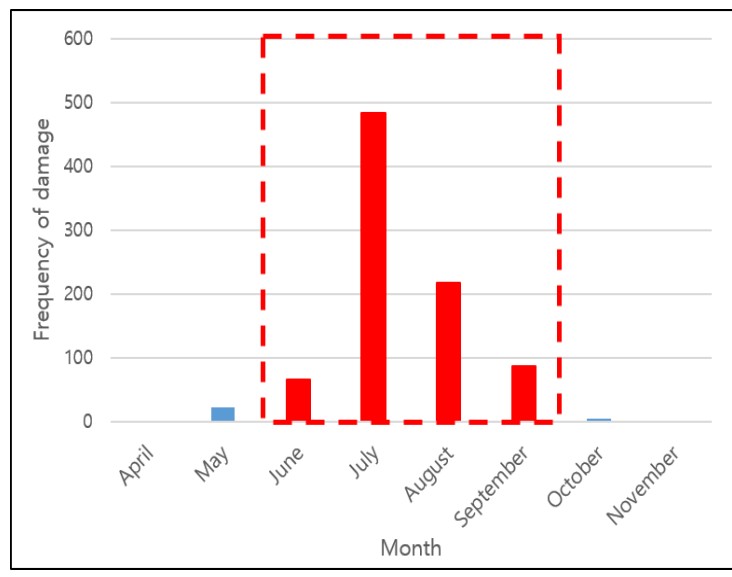

**Figure 11: Frequency of heavy rain damage in each month**



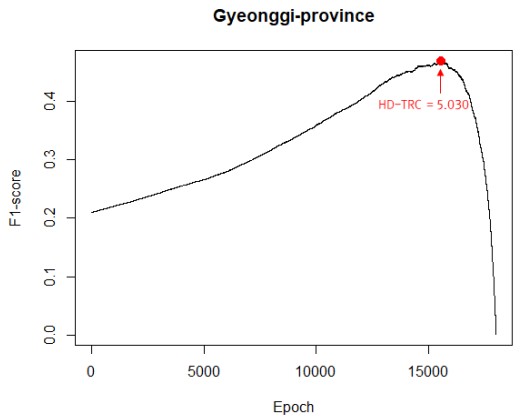

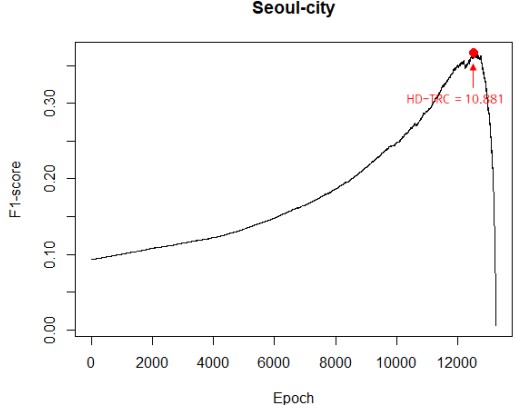

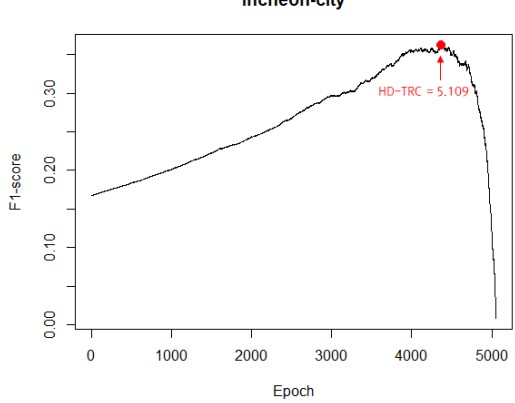

**Figure 12: Performance evaluation and determination of HD-TRC by F1-score for each region**




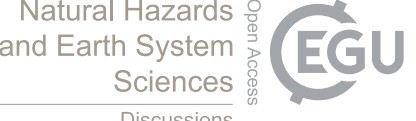

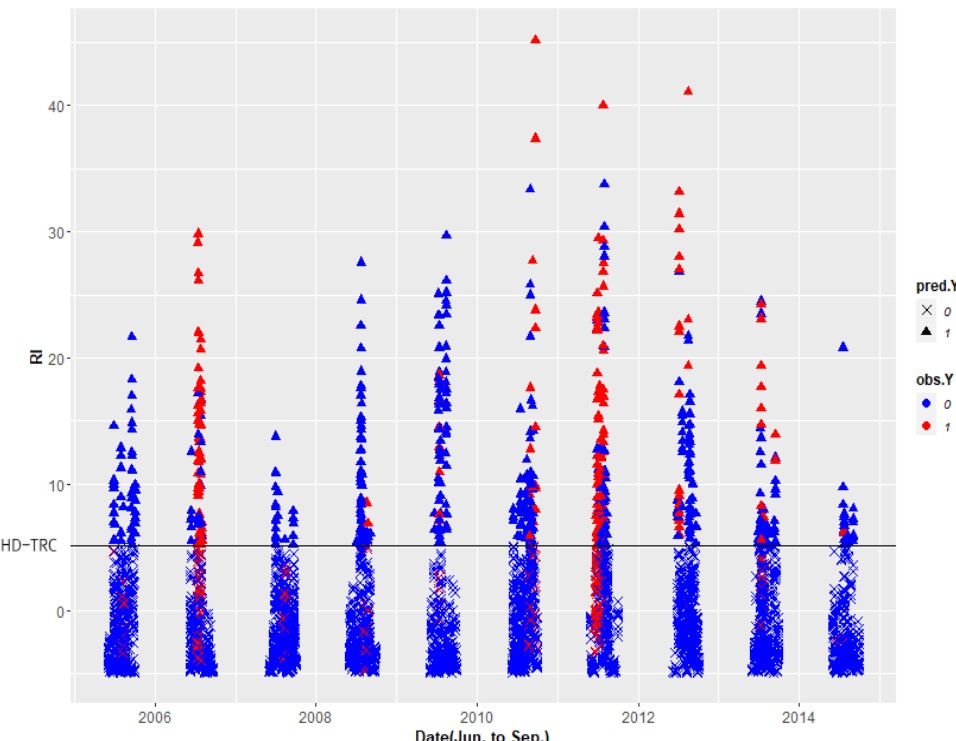

**Figure 13: Classification based on RIs from training set and the determined HD-TRC for each region : red triangle-TP, blue triangle-FP, red cross-FN, blue cross-TN (see Table 3)**





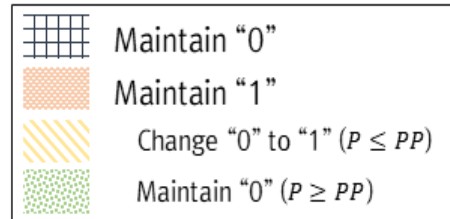

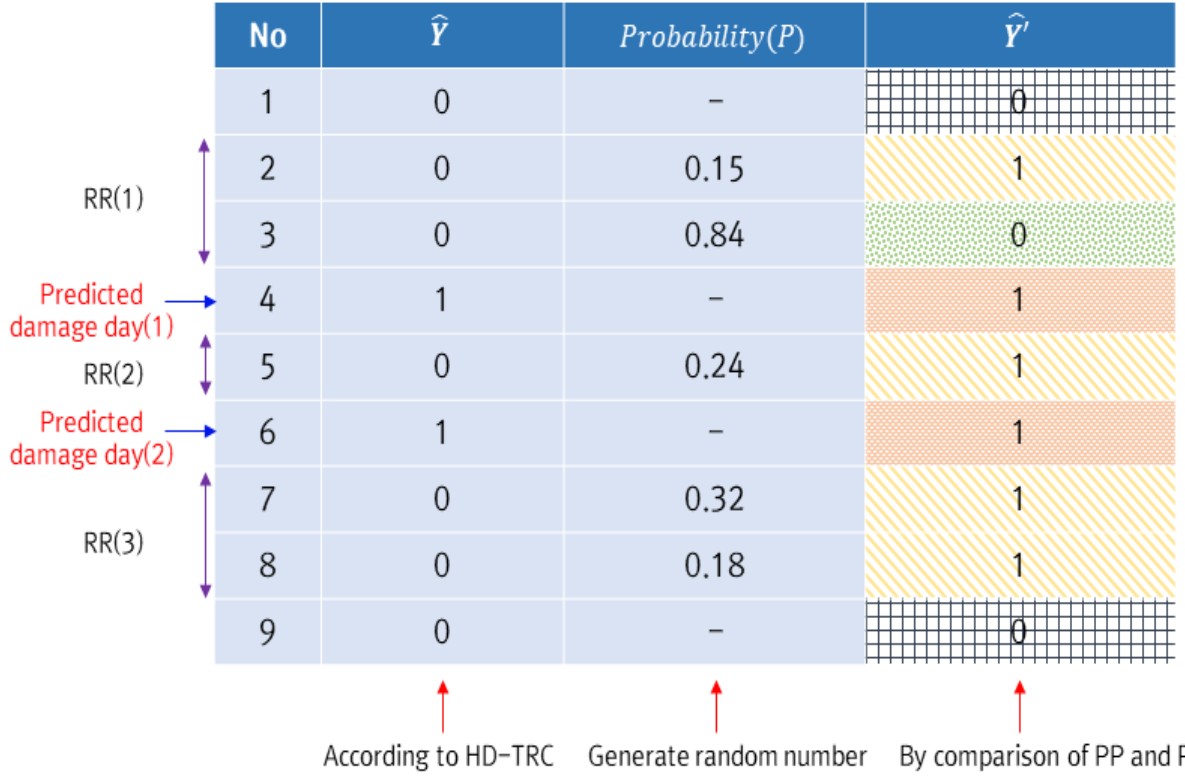

**Figure 14: An analysis example of Advanced HD-TRC using MCS with PP and RR for case 2 in Table 11**




**Table 1: Heavy rain warning criteria used in each country**

| Country | Warning criteria for heavy rain | | |
|---|---|---|---|
| **Japan** | **Heavy Rain Advisory (HRA)**<br>20mm/hr~50mm/hr(Soil Water Index 95~184) or 60mm/hr~80mm/3hr(Soil Water Index 95~184) | **Heavy Rain Warning (HRW)**<br>40mm/hr~80mm/hr(Soil Water Index 80~156) or 70mm/3hr~120mm/3hr(Soil Water Index 80~156) | |
| **Hongkong** | **Amber**<br>> 30mm/1hr | **Red**<br>> 50mm/1hr | **Black**<br>> 70mm/1hr |
| **Canada** | **Heavy Rain Warning** | | |
| | Prairie and Northern<br>> 50mm/1hr | Dry sets of British Columbia<br>> 15mm/1hr | Atlantic Region<br>> 25mm/1hr |
| **Korea** | **Heavy Rain Advisory**<br>> 60mm/3hr or > 110mm/12hr | **Heavy Rain Warning**<br>> 90mm/3hr or > 180mm/12hr | |






**Table 2: An example of SYND published by MOIS, Korea (heavy rain damage of Nowon-district in Seoul-city)**

| Damage items | | | 2018.08.26 ~ 09.01(DP)<br>Nowon-district in Seoul-city Damage<br>KRW (USD in thousand) |
|---|---|---|---|
| Casualty | | No. of persons | 0 |
| Victims | | No. of persons | 0 |
| Public facilities (Public sector) | Road | Damage KRW(1,000) | 0 |
| | Stream | Damage KRW(1,000) | 0 |
| | Small river | Damage KRW(1,000) | 0 |
| | Water supply and sewage | Damage KRW(1,000) | 0 |
| | Harbor | Damage KRW(1,000) | 0 |
| | fishing port | Damage KRW(1,000) | 0 |
| | School | Damage KRW(1,000) | 0 |
| | Railway | Damage KRW(1,000) | 0 |
| | Irrigation | Damage KRW(1,000) | 0 |
| | Erosion control | Damage KRW(1,000) | 248,640(207.2) |
| | Military | Damage KRW(1,000) | 0 |
| | Small public facilities | Damage KRW(1,000) | 0 |
| | others | Damage KRW(1,000) | 0 |
| Private facilities (Private sector) | Building | Damage KRW(1,000) | 69,300(57.75) |
| | Vessel | Damage KRW(1,000) | 0 |
| | Arable land | Damage KRW(1,000) | 0 |
| | Fence | Damage KRW(1,000) | 0 |
| | Livestock | Damage KRW(1,000) | 0 |
| | Stable | Damage KRW(1,000) | 0 |
| | Aquaculture and fish culture | Damage KRW(1,000) | 0 |
| | Fishing gear and net | Damage KRW(1,000) | 0 |
| | Vinyl house | Damage KRW(1,000) | 0 |
| | others | Damage KRW(1,000) | 0 |
| Total Damage | | KRW(1,000) | 317,940(264.95) |




**Table 3: Confusion matrix**

| Condition | | Predicted($\widehat{Y}$) | |
|---|---|---|---|
| | | 0 | 1 |
| **Observed** | 0 | TN | FP |
| (**Y**) | 1 | FN | TP |

**Table 4: Performance evaluation indicators(Powers, 2011)**

| Evaluation indicators | Equation |
|---|---|
| Accuracy | $\dfrac{TP + TN}{TP + TN + PF + FN}$ |
| Error Rate | $\dfrac{FP + FN}{TP + TN + PF + FN}$ |
| Sensitivity(recall) | $\dfrac{TP}{TP + FN}$ |
| Precision | $\dfrac{TP}{TP + FP}$ |
| Specificity | $\dfrac{TN}{TN + FP}$ |

**Table 5: Dependent and independent variables for the determination of HD-TRC**

| Variable | Description |
|---|---|
| Dependent variable | Daily damage occurrence($Y$) |
| Independent variable | Maximum rainfalls in durations of 1 to 24-hr($X_1$) |
| | Antecedent rainfalls of 1 to 7-day($X_2$) |

**Table 6: Frequency of Heavy rain Damage for the Province and Cities**

| Province and city | Frequency of heavy rain damage for each region | Average frequency of heavy rain damage for each administrative district |
|---|---|---|
| Gyeonggi-province | 616 | 19.87 (for 31 districts) |
| Seoul-city | 150 | 6 (for 25 districts) |
| Incheon-city | 121 | 12.1 (for 10 districts) |






**Table 7: No. of damage days(1) and no damage days(0) for whole years and rainy season**

| For whole years | No. of damage & no damage days | | | For rainy season | No. of damage & no damage days | |
|---|---|---|---|---|---|---|
| | 1 | 0 | | | 1 | 0 |
| Gyeonggi-province | 2,354 | 110,858 | ⇒ | Gyeonggi-province | 2,115 | 15,940 |
| Seoul-city | 740 | 90,560 | | Seoul-city | 648 | 12,647 |
| Incheon-city | 551 | 35,970 | | Incheon-city | 463 | 4,606 |

**Table 8: Cumulative variance for each rainfall variable of $X_1$ and $X_2$ in the region**

| Rainfall variable | Gyeonggi-province | Seoul-city | Incheon-city |
|---|---|---|---|
| $PC.X_1$ | 0.8009 | 0.8001 | 0.8026 |
| $PC.X_2$ | 0.9702 | 0.9708 | 0.9719 |


**Table 9: Determined HD-TRC and classification performance evaluation for each region**

| Region | HD-TRC | Sensitivity | Specificity | Precision | AUC | F1-score |
|---|---|---|---|---|---|---|
| Gyeonggi-province | 5.030 | 0.503 | 0.914 | 0.438 | 0.709 | 0.468 |
| Seoul-city | 10.881 | 0.387 | 0.963 | 0.347 | 0.675 | 0.366 |
| Incheon-city | 5.109 | 0.449 | 0.896 | 0.303 | 0.673 | 0.362 |




**Table 10: Confusion matrix of HD-TRC for each region**

| Gyeonggi-province | | Predicted ($\hat{Y}$) | |
|---|---|---|---|
| | | **0** | **1** |
| **Observed ($Y$)** | **0** | 14574 (TN) | 1366 (FP) |
| | **1** | 1051 (FN) | 1064 (TP) |

| Seoul-city | | Predicted ($\hat{Y}$) | |
|---|---|---|---|
| | | **0** | **1** |
| **Observed ($Y$)** | **0** | 12175 (TN) | 472 (FP) |
| | **1** | 397 (FN) | 251 (TP) |

| Incheon-city | | Predicted ($\hat{Y}$) | |
|---|---|---|---|
| | | **0** | **1** |
| **Observed ($Y$)** | **0** | 4128 (TN) | 478 (FP) |
| | **1** | 255 (FN) | 208 (TP) |

**Table 11: Calculations of PP and RR for each case**

| Case | Daily rainfall during DP | Total no. of days for DPs (M) | Total no. of ADs (N) | Prior Probability (PP) (N/M) | Risk Range (RR) |
|---|---|---|---|---|---|
| Case 1 | 0mm | 3,645 | 411 | 0.1128 | 1 day |
| Case 2 | 5mm | 3,645 | 1,166 | 0.3199 | 2 days |
| Case 3 | 10mm | 3,645 | 1,435 | 0.3936 | 2 days |
| Case 4 | 15mm | 3,645 | 1,656 | 0.4543 | 2 days |
| Case 5 | 20mm | 3,645 | 1,837 | 0.5040 | 3 days |




**Table 12: Result of performance evaluation in Gyeonggi-province**

| Criteria | Sensitivity | Specificity | Precision | F1-score | AUC |
|---|---|---|---|---|---|
| HRA of KMA | 6.26% | 99.94% | 66.67% | 11.44% | 53.10% |
| HD-TRC | 28.88% | 99.29% | 43.17% | 34.61% | 64.08% |
| Advanced HD-TRC 1 (Case1) | 32.49% | 99.16% | 42.06% | 36.67% | 65.77% |
| Advanced HD-TRC 2 (Case2) | 41.64% | 98.41% | 32.86% | 36.73% | 70.02% |
| Advanced HD-TRC 3 (Case3) | 45.61% | 98.16% | 31.66% | 37.36% | 71.88% |
| Advanced HD-TRC 4 (Case4) | 48.74% | 97.98% | 31.06% | 37.94% | 73.36% |
| Advanced HD-TRC 5 (Case5) | 57.04% | 97.28% | 28.16% | 37.71% | 77.16% |





**Table 13: Result of performance evaluation in Seoul-city**

| Criteria | Sensitivity | Specificity | Precision | F1-score | AUC |
|---|---|---|---|---|---|
| HRA of KMA | 14.21% | 99.84% | 32.94% | 19.86% | 57.03% |
| HD-TRC | 19.80% | 99.79% | 33.62% | 24.92% | 59.79% |
| Advanced HD-TRC 1 (Case1) | 21.83% | 99.75% | 32.33% | 26.06% | 60.79% |
| Advanced HD-TRC 2 (Case2) | 30.46% | 99.49% | 24.59% | 27.21% | 64.98% |
| Advanced HD-TRC 3 (Case3) | 34.01% | 99.41% | 23.67% | 27.92% | 66.71% |
| Advanced HD-TRC 4 (Case4) | 38.58% | 99.36% | 24.60% | 30.04% | 68.97% |
| Advanced HD-TRC 5 (Case5) | 47.72% | 99.09% | 22.07% | 30.18% | 73.40% |




**Table 14: Result of performance evaluation in Incheon-city**

| Criteria | Sensitivity | Specificity | Precision | F1-score | AUC |
|---|---|---|---|---|---|
| HRA of KMA | 6.90% | 99.97% | 73.68% | 12.61% | 53.43% |
| HD-TRC | 27.09% | 99.40% | 39.01% | 31.98% | 63.25% |
| Advanced HD-TRC 1 (Case1) | 30.05% | 99.31% | 38.13% | 33.61% | 64.68% |
| Advanced HD-TRC 2 (Case2) | 46.80% | 98.70% | 33.69% | 39.18% | 72.75% |
| Advanced HD-TRC 3 (Case3) | 49.75% | 98.43% | 30.89% | 38.11% | 74.09% |
| Advanced HD-TRC 4 (Case4) | 55.17% | 98.30% | 31.37% | 40.00% | 76.74% |
| Advanced HD-TRC 5 (Case5) | 61.08% | 97.58% | 26.27% | 36.74% | 79.33% |
