# Peer review of "Determination of Heavy Rain Damage-Triggering Rainfall Criteria Based on Data Mining"

_Natural Hazards and Earth System Sciences, 2020_

## Referee Comment (RC1) · Anonymous Referee #1 · 4 Dec 2020

Comment on Manuscript NHESS: Determination of Heavy Rain Damage-Triggering Rainfall Criteria Based on Data Mining

The manuscript deals with a methodology to determine rainfall thresholds associated with damage-triggering in Korea. The topic is relevant, and the objectives are clearly explained at the beginning. However, in my opinion the manuscript presents important problems regarding both its form and content. For example, the structure of the manuscript does not help the reader to get a clear idea of the methodology proposed, it seems more a technical report, for internal use, than a scientific paper, intended for global dissemination. While some aspects are presented in detail others are omitted or not discussed. For example it is not clear to this reviewer if maximum rainfall for the different time periods are considered with local references (for example from local IDF

series) or to some other framework, as described for rain gauge point measurements by [1] or gridded two dimensional radar derived precipitation field, as described by [2].

English language should be reviewed in depth as current problems hamper following authors descriptions. Besides there are additional issues, that should be carefully reviewed by authors. I do not intend at this stage to provide a list of specific comments, but some are: the abstract does not provide an overview, it seems part of the introduction, presenting too many details; lines 37 and 38 present inconsistent information (3.4 trilion KRW correspond to different USD amounts in each line); in line 62 authors enumerate list methods considering "statistical, empirical, and satellite" which does not make any sense; the introduction of Monte Carlo methods (line 174-175) is not done properly; Figure 15 is mentioned in line 419 but is missing, etc. Perhaps part of the problems can be solved by improving the English issues, but I think the manuscript cannot be reviewed properly in its current form.

For all the above I recommend authors to reconsider what do they want to explain, to select carefully the examples and describe in a concise way their proposal of new methodology and finally to check in depth the English version before submitting a new version of the manuscript.

References

[1] Gonzalez, S., & Bech, J. (2017). Extreme point rainfall temporal scaling: a long term (1805–2014) regional and seasonal analysis in Spain. International Journal of Climatology, 37(15), 5068-5079.

[2] Pöschmann, J. M., Kim, D., Kronenberg, R., & Bernhofer, C. (2020). An analysis on temporal scaling behavior of extreme rainfall of Germany based on radar precipitation QPE data. Natural Hazards and Earth System Sciences Discussions, 1-21.

---

## Referee Comment (RC2) · Anonymous Referee #2 · 30 Dec 2020

The overall quality of the preprint is low, it is very confused, already starting from the abstract, too detailed.

The description of method, instead is too superficial.

Many paper are mentioned in the introduction, but no results are compared and discussed in relation to the existing literature. It looks more as a technical work for consultancy, than a scientific paper.

The grammar is very low.
* * *

---

## Author Comment (AC1) · 16 Feb 2021

Comment : The manuscript deals with a methodology to determine rainfall thresholds associated with damage-triggering in Korea. The topic is relevant, and the objectives are clearly explained at the beginning.

Comment : However, in my opinion the manuscript presents important problems regarding both its form and content. For example, the structure of the manuscript does not help the reader to get a clear idea of the methodology proposed, it seems more a technical report, for internal use, than a scientific paper, intended for global dissemination.

Reply : The objective of this study is to propose a new methodology to determine

rainfall criteria or threshold of heavy rain damage which is different from the existing methodologies. For the reader, we explained a new methodology in detail which proposed in this study and also rewrote the abstract. As the reviewer mentioned, the paper can be understand as technical report or internal use, so we rewrote and rearranged the content of the manuscript. The general explanations were removed and more technically rewritten. Korea is publishing damage data every year and we used this damage data to suggest a new methodology. If other countries also have damage data (maybe the damage data can be obtained from EM-DAT), the proposed methodology can be used for the determination of criteria or threshold of heavy rain damage.

Comment : While some aspects are presented in detail others are omitted or not discussed. For example it is not clear to this reviewer if maximum rainfall for the different time periods are considered with local references (for example from local IDF series) or to some other framework, as described for rain gauge point measurements by [1] or gridded two dimensional radar derived precipitation field, as described by [2].

Reply : We rearranged the contents of a manuscript to make the balance of explanations and discussions in each section. The reference [1] studied for the characteristics of extreme rainfall events which are measured at stations using ID curve and [2] studied the cluster analysis for the examination of the rainfall similarity of regions using the gridded radar rainfall data. However, this study obtained point rainfalls and point rainfalls are transferred to mean aerial precipitations (MAP) or rainfalls which is used as independent variable to determine the criteria of heavy rain damage. Therefore, the references [1] and [2] are not directly related with this study.

Comment : English language should be reviewed in depth as current problems hamper following authors descriptions. Besides there are additional issues, that should be carefully reviewed by authors. I do not intend at this stage to provide a list of specific comments,

Reply : A native speaker corrected the manuscript for English and we reviewed the

manuscript very carefully.

Comment : the abstract does not provide an overview, it seems part of the intro- duction, presenting too many details;

Reply : We rewrote the abstract. Heavy rainfall occurs over the Korean Peninsula mainly via typhoons and localized storm events, leading to severe flooding and landslide risks. The Korean Meteorological Administration (KMA) has established equal criteria for issuing a heavy rain advisory over the peninsula, even though each region or local government has different disaster prevention conditions and rainfall and heavy rain damage characteristics. Therefore, the purpose of this study is to propose a methodology for the determination of heavy rain damage-triggering rainfall criteria (HD-TRC) that can be utilized in individual regions according to their rainfall and damage characteristics to improve a heavy rain advisory. The study regions included 66 administrative districts in Gyeonggi Province, Seoul, and Incheon, Korea. The HD-TRC were determined based on rainfall and heavy rain damage data. Therefore, data were obtained from 2005 to 2018, and the data for the flood/rainy season, which occurs from June to September, were extracted for each year. The rainfall data were obtained from the KMA, and the data for heavy rain damage that occurred during disaster periods were obtained from the Statistical Yearbook of Natural Disaster published by the Ministry of Interior and Safety. Maximum rainfall in 1- to 24-h durations and 1- to 7-d antecedent rainfall were obtained and utilized as independent variables. A principal component analysis was performed utilizing the training set, and the rain index was defined as the sum of the principal components of the independent variables. The best HD-TRC were then determined through classification performance evaluations. This study incorporated abnormal days in the analysis, which are defined as days during a disaster period with no rainfall or an insufficient amount of rainfall to cause damage. A Monte Carlo simulation was performed to consider abnormal days stochastically, and the result was defined as the advanced HD-TRC. Finally, the resultant criteria of the HD-TRC, advanced HD-TRC, and heavy rain advisory were compared based on performance evaluations utilizing the test set, which concluded that the advanced HD-TRC exhibited the best performance. Thus, its methodology could be utilized for regional heavy rain damage warnings.

Comment : lines 37 and 38 present inconsistent information (3.4 trilion KRW correspond to different USD amounts in each line);

Reply : We corrected it.

Comment : in line 62 authors enumerate list methods considering "statistical, empirical, and satellite" which does not make any sense;

Reply : We rewrote the introduction, and deleted the corresponding contents.

Comment : the introduction of Monte Carlo methods (line 174-175) is not done properly;

Reply : We just wrote the general MCM and how to use MCM in this study.

Comment : Figure 15 is mentioned in line 419 but is missing, etc.

Reply : Figure 15 is a typo and we corrected it.

Comment : Perhaps part of the problems can be solved by improving the English issues, but I think the manuscript cannot be reviewed properly in its current form.

Reply : A native speaker corrected the manuscript for English and we reviewed the manuscript very carefully.

Comment : For all the above I recommend authors to reconsider what do they want to explain, to select carefully the examples and describe in a concise way their proposal of new methodology and finally to check in depth the English version before submitting a new version of the manuscript.

Reply : We revised the manuscript carefully for writing and explaining a methodology in concise way and also a native speaker checked English. Thank you very much for
very valuable comments of the reviewer.

---

## Author Comment (AC2) · 16 Feb 2021

Comment : The overall quality of the preprint is low, it is very confused, already starting from the abstract, too detailed. The description of method, instead is too superficial.

Reply : We rewrote the abstract. Heavy rainfall occurs over the Korean Peninsula mainly via typhoons and localized storm events, leading to severe flooding and landslide risks. The Korean Meteorological Administration (KMA) has established equal criteria for issuing a heavy rain advisory over the peninsula, even though each region or local government has different disaster prevention conditions and rainfall and heavy rain damage characteristics. Therefore, the purpose of this study is to propose a methodology for the determination of heavy rain damage-triggering rainfall criteria

(HD-TRC) that can be utilized in individual regions according to their rainfall and damage characteristics to improve a heavy rain advisory. The study regions included 66 administrative districts in Gyeonggi Province, Seoul, and Incheon, Korea. The HD-TRC were determined based on rainfall and heavy rain damage data. Therefore, data were obtained from 2005 to 2018, and the data for the flood/rainy season, which occurs from June to September, were extracted for each year. The rainfall data were obtained from the KMA, and the data for heavy rain damage that occurred during disaster periods were obtained from the Statistical Yearbook of Natural Disaster published by the Ministry of Interior and Safety. Maximum rainfall in 1- to 24-h durations and 1- to 7-d antecedent rainfall were obtained and utilized as independent variables. A principal component analysis was performed utilizing the training set, and the rain index was defined as the sum of the principal components of the independent variables. The best HD-TRC were then determined through classification performance evaluations. This study incorporated abnormal days in the analysis, which are defined as days during a disaster period with no rainfall or an insufficient amount of rainfall to cause damage. A Monte Carlo simulation was performed to consider abnormal days stochastically, and the result was defined as the advanced HD-TRC. Finally, the resultant criteria of the HD-TRC, advanced HD-TRC, and heavy rain advisory were compared based on performance evaluations utilizing the test set, which concluded that the advanced HD-TRC exhibited the best performance. Thus, its methodology could be utilized for regional heavy rain damage warnings.

Comment : Many paper are mentioned in the introduction, but no results are compared and dis- cussed in relation to the existing literature.

Reply : We rewrote introduction part. The criteria of heavy rain warning is related with three main types of flood, landslide, and heavy rain warnings. Flood warnings can be issued based on threshold values obtained from flood discharge forecasting, landslide warnings can be issued based on rainfall (ID) curves, and heavy rain warnings (HRWs) can be issued based on rainfall events. Flood forecasting is performed utilizing rainfallrunoff models, and warning criteria have been proposed based on forecasted threshold discharges (Beguería et al., 2006; Montesarchio et al., 2009; Alfieri et al., 2015; Forestieri et al., 2016; Miao et al., 2016; Sieq et al., 2017; Zhai et al., 2018; Sairam, 2019); however, the rainfall-runoff model can be used just for a specific river basin and waring is also issued for a basin. In previous landslide criteria determination studies, the landslide damage data were collected, and then the damage-triggering rainfall criteria were determined based on the ID curve, which was constructed from the damage data (Glade et al., 2000; Cannon et al., 2008; Dahal and Hasegawa, 2008; Saito et al., 2010; Piciullo et al., 2016; Lee et al., 2017). Because landslides are influenced by soil moisture content, antecedent rainfall should be considered; however, most previous studies have only considered the ID curve to determine the criteria. A previous study was performed to determine landslide criteria utilizing antecedent rainfall (Glade et al., 2000). Antecedent rainfall is an important factor in heavy rain damage studies because it affects soil conditions and flooding. Most previous studies for improving the heavy rain advisory (HRA) criteria only included events in which damage occurred, and the criteria were determined as the minimum rainfall among the included events (Kim et al., 2011; Montesarchio et al., 2015; Lopez et al., 2017; Cho et al., 2018). However, many events can have occurred in which no damage was sustained, even though the rainfall events were greater than the determined minimum rainfall in this case; therefore, the reliability of the warning system will be degraded. Additionally, because the aforementioned studies utilized the entire dataset to determine the criteria, there were no data for criteria verification. The purpose of this study is to propose a methodology for the determination of the three criteria, heavy rain damage-triggering rainfall criteria (HD-TRC), advanced HD-TRC, and HRA issued by the Korean Meteorological Administration (KMA). Subsequently, the results of these three criteria are compared to determine the best criteria. The training data set from 2005–2014 data was utilized to obtain the HD-TRC, and the test data set from 2015–2018 was utilized to evaluate the criteria of the HD-TRC, advanced HD-TRC, and HRA. The analysis for determining the best criteria was performed via the following steps: (1) Maximum rainfalls in durations of 1 to 24-hr (X1) and antecedent rainfalls of 1 to 7-day (X2) are obtained and used as independent variables. Heavy rain damage data are divided into damage day ("1") and no damage day ("0") used as dependent variables (Y). Principal component analysis (PCA) is performed and PCs (principal components) are obtained as and for independent variables. Then Risk Index (RI) is defined as X1 + X2 and RIs become the candidates for HD-TRC. The predicted damage() is obtained based on RIs and confusion matrix is constructed then the best HD-TRC is determined through the evaluation of classification performance. (2) However, 'abnormal days' (ADs) in a DP that the damage is occurred exists. The ADs mean the days which we do not have rainfall or have small rainfall amount during DP. Say, ADs have too small rainfall to damage even during DP. The ADs are defined as days below rainfall of 20mm and 5 cases of ADs are also defined as 0, 0∼5, 0∼10, 0∼15, and 0∼20mm in this study. We count total days in all the DPs and in ADs for a case. The ratio of ADs to total days during DPs could be the occurrence probability or prior probability (PP) of ADs for a case and 5 PPs are obtained. Also, the average AD for each case can be obtained and defined as risk range (RR). Then we define Advanced HD-TRC using MCS (Monte Carlo Simulation) linked with PP, RR, and HD-TRC. Therefore, HD-TRC is determined based on RI and Advanced HD-TRC for each case based on MCS. Finally, three criteria of HD-TRC, Advanced HD-TRC, and HRA are compared based on performance evaluation by test set for the determination of the best criterion.

Comment : It looks more as a technical work for consultancy, than a scientific paper. The grammar is very low.

Reply : The objective of this study is to propose a new methodology to determine rainfall criteria or threshold of heavy rain damage which is different from the existing methodologies. For the reader, we explained a new methodology in detail which proposed in this study and also rewrote the abstract. As the reviewer mentioned, the paper can be understand as technical report or internal use, so we rewrote and rearranged the content of the manuscript. The general explanations were removed and more technically rewritten. Korea is publishing damage data every year and we used this damage data to suggest a new methodology. A native speaker corrected the manuscript for English and we reviewed the manuscript very carefully.
* * *